# Favipiravir in the Treatment of Outpatient COVID-19: A Multicenter, Randomized, Triple-Blind, Placebo-Controlled Clinical Trial

Atefeh Vaezi [1], Mehrzad Salmasi [2], Forogh Soltaninejad [3], Mehrdad Salahi [4,*], Shaghayegh Haghjooy Javanmard [5] and Babak Amra [3]

1   Cancer Prevention Research Center, Isfahan University of Medical Sciences, Isfahan 8174673461, Iran
2   Department of Internal Medicine, School of Medicine, Isfahan University of Medical Sciences, Isfahan 8174673461, Iran
3   Bamdad Respiratory and Sleep Research Center, Department of Internal Medicine, School of Medicine, Isfahan University of Medical Sciences, Isfahan 8174673461, Iran
4   Department of Infectious Disease, School of Medicine, Isfahan University of Medical Sciences, Isfahan 8174673461, Iran
5   Department of Physiology, Applied Physiology Research Center, Cardiovascular Research Institute, Isfahan University of Medical Sciences, Isfahan 8174673461, Iran
*   Correspondence: salahimehrdad@yahoo.com

**Highlights:**

**What are the main findings?**

- Favipiravir, an RNA-dependent RNA polymerase inhibitor, shows no benefit in preventing the hospitalization of mild to moderate COVID-19 patients.

**What is the implication of the main finding?**

- Our results may inform decisions on the exclusion of Favipiravir from mild to moderate COVID-19 treatment guidelines.

**Abstract:** Background: Finding effective outpatient treatments to prevent COVID-19 progression and hospitalization is necessary and is helpful in managing limited hospital resources. Repurposing previously existing treatments is highly desirable. In this study, we evaluate the efficacy of Favipiravir in the prevention of hospitalization in symptomatic COVID-19 patients who were not eligible for hospitalization. Methods: This study was a triple-blind randomized controlled trial conducted between 5 December 2020 and 31 March 2021 in three outpatient centers in Isfahan, Iran. Patients in the intervention group received Favipiravir 1600 mg daily for five days, and the control group received a placebo. Our primary outcome was the proportion of hospitalized participants from day 0 to day 28. The outcome was assessed on days 3, 7, 14, 21, and 28 through phone calls. Results: Seventy-seven patients were randomly allocated to Favipiravir and placebo groups. There was no significant difference between groups considering baseline characteristics. During the study period, 10.5% of patients in the Favipiravir group and 5.1% of patients in the placebo group were hospitalized, but there was no significant difference between them (*p*-value = 0.3). No adverse event was reported in the treatment group. Conclusions: Our study shows that Favipiravir did not reduce the hospitalization rate of mild to moderate COVID-19 patients in outpatient settings.

**Keywords:** efficacy; Favipiravir; mild COVID-19; outpatient; SARS-CoV-2

## 1. Introduction

SARS-CoV-2 infected more than 635 million people as of October 2022, from which 6.5 million people died [1]. The pooled prevalence of mortality of COVID-19 patients is 17% among hospitalized patients [2]. COVID-19 has imposed high direct and indirect costs

on healthcare systems. The average medical cost of COVID-19 per person was estimated at 3755 USD in Iran and 3045 USD in the United States of America [3]. Hence, from a public health perspective, one important measure is the prevention of hospitalization.

Antiviral prescription in the early phase of the infection can decrease the duration of viral shedding and the intensity of the immune response [4]. Although the best timing of antiviral therapy is still unknown [5], the ideal time might be before the spread of the infection from the upper to the lower respiratory tract and before the beginning of the severe inflammatory reaction [6]. Therefore, the management of COVID-19 in outpatient settings as early as possible may be a critical first step to slow down the pandemic curve, lower the pressure on hospitals, and even stop the pandemic more effectively.

Different antivirals have been studied to treat COVID-19, including Ivermectin, Remdesivir, Lopinavir, and Favipiravir [7–10]. Favipiravir, an antiviral drug, is an RNA-dependent RNA polymerase inhibitor and has proven efficacy against influenza viruses [11,12]. Studies focusing on the therapeutic effect of Favipiravir consider different outcomes, including clinical improvement, time to virus clearance, hospitalization, ICU admission, and mortality; existing studies have reported controversial results [13,14].

In most recent studies, Favipiravir was prescribed to hospitalized patients. Given the promising yet contradictory results of the therapeutic effect of Favipiravir on COVID-19, we decided to evaluate the efficacy of Favipiravir in the prevention of the hospitalization of symptomatic mild to moderate COVID-19 patients, with no criteria for hospitalization, in outpatient settings.

## 2. Materials and Methods

### 2.1. Study Design

This triple-blind randomized controlled trial was conducted in three outpatient centers in Isfahan, Iran, between 5 December 2020 and 31 March 2021. This study was conducted following the Good Practice Guideline. The study protocol was approved by the Ethics Committee of the Isfahan University of Medical Sciences (IR.MUI.MED.REC.1399.780). Patients were informed about the study protocol and were asked to sign an informed consent before participation. The study protocol was registered in the Iranian Registry of Clinical Trials on 2 December 2020 (IRCT20171219037964N3). We follow the Consolidated Standards for Reporting Trials (CONSORT 2010) guideline in reporting the results [15].

### 2.2. Participants

Confirmed cases of COVID-19 patients were eligible for this trial if they were 18 years and older, referred to any of the outpatient settings of the study in the first seven days of their symptoms, and had an oxygen saturation of 93% or more on room air. Cases of severe COVID-19 who had criteria for hospitalization, pregnant or breastfeeding women, patients with renal or hepatic failure, patients who were under treatment with corticosteroids, patients with a previous history of COVID-19 infection, those who received any antiviral medication within two weeks before the first dose of the study treatment, and those with a possible allergic reaction to Favipiravir were excluded.

### 2.3. Randomization and Blinding

Participants were randomly assigned in a 1:1 ratio to oral Favipiravir or placebo. The same pharmacological factory made the placebo and Favipiravir. The placebo was the same as Favipiravir in size, shape, and color. An independent staff packed Favipiravir and placebo in identical boxes and labeled them A or B. The random string was generated using "Randomization.Org" by the project manager. The participants, physicians, outcome evaluator, data analyzer, and the project manager were unaware of the treatment assignments.

### 2.4. Procedure, Intervention, and Outcome

Participants were randomized to receive Favipiravir or placebo. Patients in the intervention group received Favipiravir 1600 mg daily for five days (Cytovex, 200 mg, Abidi

pharmacological Inc., Tehran, Iran). Patients in the control group received placebo with the same instructions as the intervention group.

Demographic data such as age, gender, job, education, height, and weight, date of the first symptom, presenting symptoms (fever, cough, headache, dyspnea, gastrointestinal symptoms, sore throat, rhinorrhea, chills, body pain, loss of appetite, loss of smell or taste, and fatigue), and history of underlying comorbidities (cardiovascular disease, diabetes, hypertension, asthma, and cancer) were documented at the first visit. All patients, even those who did not follow the instructions, were followed up on the 3rd, 7th, 14th, 21st, and 28th day via phone call.

The primary outcome of this study was hospitalization. The secondary outcome was adverse events (AEs), which was reported through Common Terminology Criteria for Adverse Events (CTCAE) version 4.0. The AEs and patients' compliance were assessed through the first and second follow-up calls. In the case of incompliance, the reason was asked and recorded.

### 2.5. Data Analysis

Data were analyzed using SPSS version 25 (IBM Corp., Armonk, NY, USA). Mean and standard deviation were used to describe continuous variables. Frequency was used to present categorical variables. The independent sample *t*-test and chi-square were used to compare continuous and categorical variables between the two groups. The survival analysis was used to compare the time to primary outcome between groups. The intention to treat (ITT) method was used for analysis. The significant level was considered at 0.05.

## 3. Results

One hundred and twelve patients were assessed for eligibility, and seventy-seven were included in this study. Thirty-eight patients were randomly allocated to the Favipiravir group and thirty-nine to the placebo group (Figure 1).

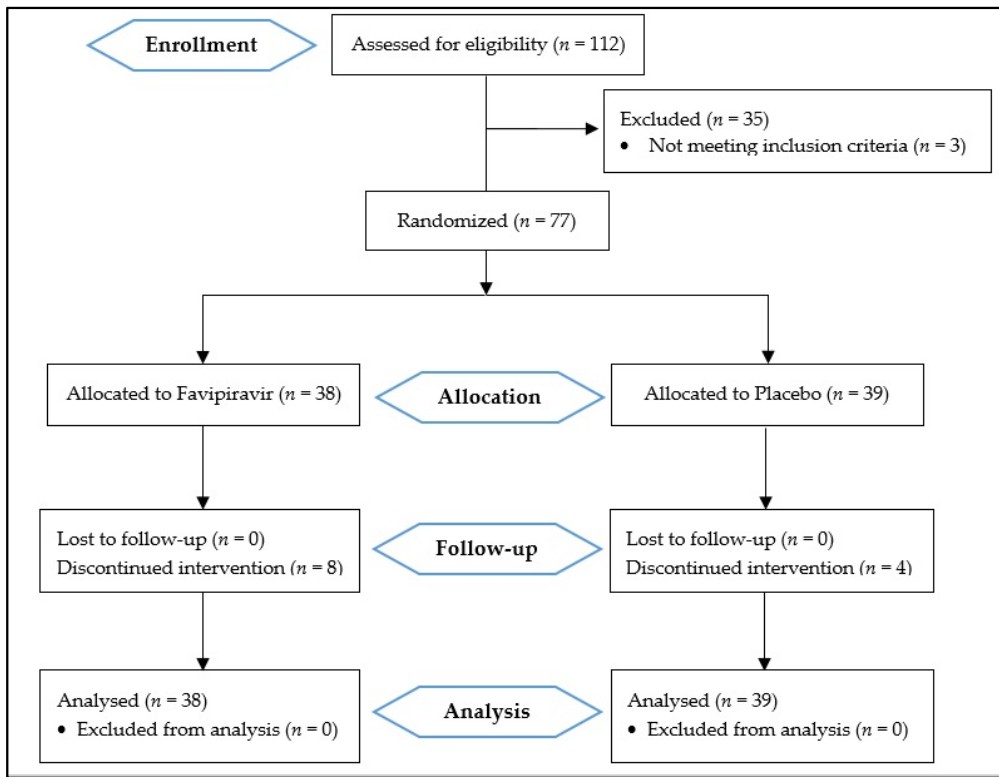

**Figure 1.** Consolidated standards of reporting trials (CONSORT 2010) flow diagram.

Eight patients in the treatment group and four in the placebo group did not complete the treatment. In the Favipiravir group, three patients discontinued the treatment because they believed that they did not need any medication. One patient declared that because the pills were not labeled, she could not trust them and decided not to use them. Others said that the number of tablets was so many, and it was hard for them to consume so many tablets. The reasons for discontinuation in the control group were the number of tablets in two patients, an AE in one patient, and not specified in the other one. The follow-up was completed for all patients regardless of their compliance.

The demographic characteristics of the patients are shown in Table 1. The mean ± SD age of patients was 41.1 ± 12.88 and 40.9 ± 12.93 years in the treatment and placebo groups, respectively (*p*-value = 0.9). There was no significant difference between groups considering gender, job, education, previous comorbidities, and body mass index (BMI). The most prevalent presenting symptom in the Favipiravir and placebo group was fever (76.3%) and fatigue (66.7%), respectively. The frequency of signs and symptoms was balanced between the two groups. The mean ± SD of time between the symptom initiation and entering the study was also the same between the two groups (3.9 ± 1.49 vs. 3.8 ± 1.45 days in the treatment and placebo groups, respectively; *p*-value = 0.87).

**Table 1.** Demographic and clinical characteristics of the patients by treatment groups.

| Characteristics | Favipiravir (*n* = 38) | Placebo (*n* = 39) | *p*-Value |
|---|---|---|---|
| Age, year Mean (±SD) | 41.1 (±12.88) | 40.9 (±12.93) | 0.9 |
| Gender | | | 0.3 |
|     Male | 19 (50.0) | 24 (61.5) | |
|     Female | 19 (50.0) | 15 (38.5) | |
| Job | | | 0.8 |
|     Self-employed | 15 (39.5) | 13 (33.3) | |
|     Housekeeper or retired | 11 (28.9) | 12 (30.8) | |
|     Students | 2 (5.3) | 1 (2.6) | |
|     Employee | 10 (26.3) | 13 (33.3) | |
| Education | | | 0.9 |
|     Diploma and lower | 18 (47.4) | 18 (46.2) | |
|     Higher education | 20 (52.6) | 21 (53.8) | |
| Any comorbidities | 5 (13.2) | 6 (16.2) | 0.7 |
| BMI ≥ 25 | 26 (68.4) | 22 (57.9) | 0.3 |
| Signs and symptoms | | | |
|     Fever | 29 (76.3) | 24 (61.5) | 0.1 |
|     Cough | 21 (55.3) | 21 (53.8) | 0.9 |
|     Headache | 21 (55.3) | 21 (53.8) | 0.9 |
|     Dyspnea | 11 (28.9) | 9 (23.1) | 0.5 |
|     Gastrointestinal | 13 (34.2) | 13 (33.3) | 0.9 |
|     Sore throat | 11 (28.9) | 13 (33.3) | 0.6 |
|     Rhinorrhea | 16 (42.1) | 11 (28.2) | 0.2 |
|     Chills | 21 (55.3) | 19 (48.7) | 0.5 |
|     Body pain | 27 (71.1) | 25 (64.1) | 0.5 |
|     Loss of appetite | 15 (39.5) | 10 (25.6) | 0.1 |
|     Loss of smell or taste | 8 (21.1) | 10 (25.6) | 0.6 |
|     Fatigue | 28 (73.7) | 26 (66.7) | 0.5 |
| Mean days (±SD) between symptom initiation and entering the study | 3.9 (±1.49) | 3.8 (±1.45) | 0.87 |

Data were reported as *n* (%), unless otherwise mentioned. Data were analyzed using chi-square, otherwise mentioned. BMI: body mass index; SD: standard deviation.

The primary outcome happened in four patients (10.5%) in the treatment group and two patients (5.1%) in the control group through the 28 days of follow-up. The chi-square test result shows no difference in the proportion of hospitalization between the two groups

(*p*-value = 0.3) (Table 2). Also, the results of the Kaplan–Meier analysis found no difference in the time to hospitalization between the two groups (log-rank *p*-value = 0.307) (Figure 2).

**Table 2.** Primary and secondary outcomes in the study groups.

| Outcome | Favipiravir (*n* = 38) | Placebo (*n* = 39) | *p*-Value |
|---|---|---|---|
| Primary outcome Hospitalization | 4 (10.5) | 2 (5.1) | 0.3 * |
| Secondary outcome Adverse events | 0 | 1 (2.6) | - |

Data are reported as *n* (%). * Data were analyzed using chi-square.

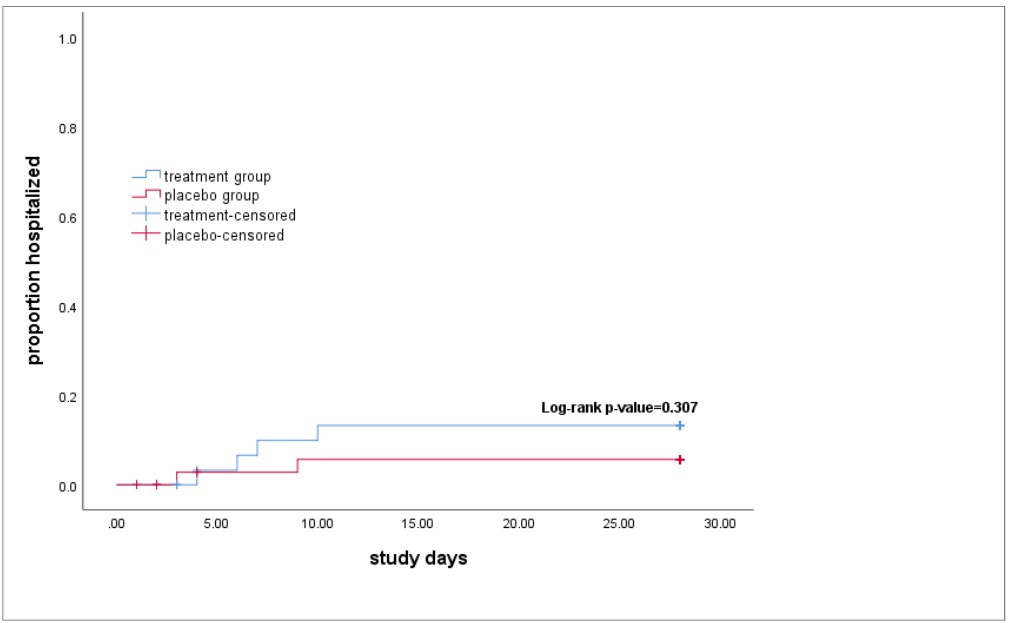

**Figure 2.** Time to the hospitalization in study groups. The *p*-value calculated by the log-rank test in Kaplan–Meier analysis showed no difference in the time to hospitalization between the treatment and the placebo groups.

No major AEs were reported during the follow-up. Only one patient in the placebo group complained about abdominal discomfort that improved after discontinuing the intervention.

## 4. Discussion

The results of our triple-blinded randomized controlled trial show no benefit of Favipiravir in preventing the hospitalization of mild to moderate COVID-19 patients in 28 days of follow-up. The results of our study were in line with the results of most previous studies on the lack of benefits of Favipiravir on COVID-19 patients' outcomes. Several trials focused on the effect of Favipiravir on the treatment of COVID-19 patients, mostly hospitalized and a few in outpatient settings.

Hassaniazad et al. [16] and Khamis et al. [17] studied the efficacy of Favipiravir combined with interferon beta; the first study administered Favipiravir for five days and the second study for a maximum of ten days. Both studies showed no benefit from combined Favipiravir regimen on the length of hospital stay, viral clearance, ICU admission, and mortality. Another study compared the efficacy of Favipiravir with hydroxychloroquine and standard care in 150 mild to moderate COVID-19 patients and revealed no association between Favipiravir and clinical improvement. In that study, the time to viral clearance was shorter in the Favipiravir and hydroxychloroquine groups, but it was not statistically significant [18].

On the other hand, a multicentric, randomized clinical trial by Ivashchenko A.A. et al. found promising effects of Favipiravir in hospitalized COVID-19 patients. In that study, the viral clearance rate on day five was higher in those who received Favipiravir (1600 mg on day one/600 mg on days 2–14 or 1800 mg on day one/800 mg on days 2–14) than in those who received standard care. However, the rate of viral clearance on day ten and the rate of improvement of chest CT scan on day 15 were the same in both groups [19]. The results of another multicentric randomized controlled trial by Sirijatuphat R. et al. that administered Favipiravir for a minimum of five days and a maximum of 14 days showed that the early start of Favipiravir, before the onset of pneumonia, is associated with faster clinical improvements [20].

To the best of our knowledge, one study has considered hospitalization as an outcome of the effect of Favipiravir [21]: Bosaeed et al. studied the efficacy of Favipiravir compared to placebo in adults with mild COVID-19. In this study, participants received Favipiravir for a total duration of five to seven days. The results show that Favipiravir was not associated with any of the outcomes, including viral clearance, clinical improvement, or hospitalization through 15 days of follow-up. The hospitalization rate in the treatment group was more than three times the rate in the control group.

This discrepancy in results is also apparent in systematic reviews. The results of a systematic review by Deng W. et al. showed that Favipiravir reduces the virus clearance time and hospitalization period in mild to moderate COVID-19 patients but not in severe cases [22]. Another systematic review and meta-analysis by Shrestha D.B. et al. revealed that Favipiravir is not associated with earlier virus clearance or needed oxygen support compared to standard care [23]. The results of a recent meta-analysis by Kow C.S. et al. showed that Favipiravir has no effect on the prevention of mortality in COVID-19 patients. In this meta-analysis, different severities of disease were included [24].

The contradictory results of Favipiravir studies could be due to different factors, including the lack of one definition for the severity of the disease, the study design, the dosage and duration of therapy, the situation of the outbreak in the society, and various characteristics of the study sample.

Based on a study by Reddy P.K. et al. on 1083 mild-moderate COVID-19 patients, AEs were reported in 12% of patients, of which 11% were mild [25]. The rate of AEs of Favipiravir and withdrawal of Favipiravir due to AEs in another study by Joshi S. et al. was 0.4% and 1.1%, respectively. AEs include gastrointestinal problems, elevation of uric acid, blood triglyceride, and liver function parameters, decreased neutrophil count, and psychological symptoms [26]. In our study, we did not document any AEs, but in terms of compliance, the relatively high number of tablets was a determinant of withdrawal of the intervention.

Our study has some limitations. This study would benefit from more objective outcomes, such as improvement in oxygen saturation or viral clearance. Considering the outpatient setting of this study, patients were not supposed to return to the clinics, so we were limited in choosing objective outcomes. Another limitation was the sample size of our study along with noncompliance of patients mainly due to the high number of tablets they were supposed to consume. Hence, we suggest future studies should be conducted with a larger sample size. Moreover, the lack of consensus on the doses and duration of Favipiravir indicates an urgent need for more studies.

## 5. Conclusions

In conclusion, our study found that Favipiravir has no efficacy in reducing the hospitalization of mild to moderate COVID-19 patients.

**Author Contributions:** Conceptualization: A.V., M.S. (Mehrdad Salahi), S.H.J. and B.A.; Formal analysis: A.V.; Investigation: M.S. (Mehrzad Salmasi), F.S., M.S. (Mehrdad Salahi) and B.A.; Methodology: A.V. and F.S.; Project administration: A.V.; Supervision: M.S. (Mehrdad Salahi), S.H.J. and B.A.; Validation: M.S. (Mehrzad Salmasi); Writing—original draft: A.V., M.S. (Mehrzad Salmasi) and S.H.J.; Writing—review and editing: A.V., M.S. (Mehrzad Salmasi), F.S., M.S. (Mehrdad Salahi), S.H.J. and B.A. All authors have read and agreed to the published version of the manuscript.

**Funding:** This study was supported by the Isfahan University of Medical Sciences.

**Institutional Review Board Statement:** This study was conducted following the Good Practice Guideline. The study protocol was approved by the Ethics Committee of Isfahan University of Medical Sciences (IR.MUI.MED.REC.1399.780).

**Informed Consent Statement:** Patients were informed about the study protocol and objectives and were asked to sign an informed consent before participation. All data were managed, analyzed, and reported anonymously. Medications were provided by the research team and imposed no costs on patients.

**Data Availability Statement:** The data analyzed during the current study are available from the corresponding author upon reasonable request.

**Acknowledgments:** The authors would like to thank all patients who attended this study and patiently followed our study protocol.

**Conflicts of Interest:** The authors declare that they have no conflict of interest.

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
