# Peer review of "Favipiravir in the Treatment of Outpatient COVID-19: A Multicenter, Randomized, Triple-Blind, Placebo-Controlled Clinical Trial"

_arm, doi:10.3390/arm91010004_

Round 1

Reviewer 1 Report

REVIEW REPORT

This study attempts to answer the controversial question of the efficacy of Favipiravir, which is an RNA-dependent RNA polymerase inhibitor, with a proven effect on the influenza virus.

A robust methodology using a triple-blind randomized controlled trial, which gives us higher Level 2 evidence on the question of the benefit of Favipiravir. The study has been appropriately approved by Ethics Committee and registered with the trial registry authority (IRCT20171219037964N3). CONSORT and GCP guidelines were also followed. Patients were followed up on the 3rd, 7th, 14th, 21st, and 28th days via phone call. Good biostatistical methods were used in the analysis of their data. Triple-blinding, subject matching, a larger sample size of 77, and minimal trial attrition, increase the validity of their research. The language of the paper is also flawless. This a praiseworthy and good-quality research paper by the authors.

The ethical issue is inherent in randomized controlled trials because the control arm of the trial received only Placebo. These patients would be at a disadvantage and more risk of developing Covid-19 hence the Ethical problem. There are some errors consistency of language style, I have offered corrections. The discussion section has several instances, where the authors refer to previous research as “another study”, or “a study”. The authors must be more specific, and I have offered corrections.

Suggestions

Line 52: “including Ivermectine,” correct spelling is “Ivermectin”

Line 105: change ‘third’ to 3rd – to maintain uniformity of style in your work

Line 121: change 77 to “seventy-seven” – to maintain uniformity of style in your work

Line 177-179: Delete lines “Authors should……. may also be highlighted”. These lines are included by mistake.

Line 194: “On the other hand, a randomized clinical trial finds Promising effects of Favipiravir” change this to be more specific like this- “On the other hand, a multicentric, randomized clinical trial by Ivashchenko A. A. et. al. found the benefit of Favipiravir”

Line 194: “Promising” please use “promising”

Line 198: “Results of another study pointed out that the early start of Favipiravir,” please change this to be more specific like this- “Results of another a multicentric, randomized control study by this Sirijatuphat R. et. al.  pointed out that the early start of  Favipiravir,”

Line 201-202: “To the best of our knowledge, one study considers hospitalization as an outcome of the effect of Favipiravir.” Which study?? There is no citation or names. Please clarify this line or delete these lines.

Line 207-208 “The results of the systematic review show that Favipiravir reduces the virus clearance time and hospitalization” please change this to be more specific like this- “The results of an expert review by Deng W. et. al. show that Favipiravir”

Line 209-210: “Another systematic review reveals that favipiravir” please change this to be more specific like this- “Another systematic review and meta-analysis by Shrestha D.B. et. al. reveals that favipiravir”

Line 211-12: “results of a recent meta-analysis show that Favipiravir has no effect on the prevention” please change this to be more specific like this- “results of a recent meta-analysis by Kow C.S. et. al. show that Favipiravir has no effect on the prevention”

Line 217: “Based on a study on 1083 mild-moderate Covid-19 patients reported a rate of 12%” please change this to be more specific like this- “Based on a study by Reddy P.K. et. al. on 1083 mild-moderate Covid-19 patients reported a rate of 12%”

Line 218-219: “withdrawal of Favipiravir due to AEs in another study was 0.4% and 1.1%, respectively.” please change this to be more specific like this- “withdrawal of Favipiravir due to AEs in another research by Joshi S. et. al. was 0.4% and 1.1%, respectively.”

Author Response

Dear Reviewer, thanks for your comments and suggestions.

we have incorporated all the suggestions (in the attachment).

Regards

Reviewer 2 Report

1. The mention of other studies of Favipiravir, specifically in COVID-19 patients, should be saved for the discussion rather than the introduction. 

2. Why was it determined to use favipiravir for 5 days, and how does this treatment duration compare to other studies?

3. In line 98, it is unclear what "exact instructions" means.

4. Table at line 165 needs a title.

5. In line 228, it is unclear what "almost low" means.

6. There are typos in the references and missing punctuation throughout.

7. The structure of the discussion is hard to follow because every paragraph jumps back and forth between studies that showed benefits and studies that did not.

Author Response

Dear Reviewer

Thanks a lot for your comments. Our point-by-point responses are presented in the attached file.

Regards

Round 2

Reviewer 1 Report

REVIEW REPORT-2

The authors have meticulously corrected and revised the manuscript and incorporated all the suggestions. The manuscript can be accepted in its present form. The authors are congratulated for producing a high-quality, original, and concise synthesis.

Reviewer 2 Report

Overall, well done. Thank you for being receptive to the feedback.